# The Predicting Role of the Neutrophil-to-Lymphocyte Ratio for the Tumor Grade and Prognosis in Pancreatic Neuroendocrine Tumors

**DOI:** 10.3390/diagnostics12030737

**Published:** 2022-03-18

**Authors:** Dongwook Oh, Jung-Soo Pyo, Kwang Hyun Chung, Byoung Kwan Son

**Affiliations:** 1Department of Gastroenterology, Asan Medical Center, University of Ulsan College of Medicine, Seoul 05505, Korea; dongwook.oh1@gmail.com; 2Department of Pathology, Uijeongbu Eulji Medical Center, Eulji University School of Medicine, Uijeongbu-si 11759, Korea; jspyo@eulji.ac.kr; 3Department of Internal Medicine, Uijeongbu Eulji Medical Center, Eulji University School of Medicine, Uijeongbu-si 11759, Korea; kh.chung@eulji.ac.kr

**Keywords:** pancreatic neuroendocrine tumor, inflammation mediators, biomarkers, tumor grade, prognosis, meta-analysis

## Abstract

This study aims to investigate the prognostic role of the neutrophil-to-lymphocyte ratio (NLR) in pancreatic neuroendocrine tumors (PNETs) using meta-analysis. This study evaluates the correlation between the NLR and the prognosis in PNETs from nine eligible studies. In addition, a subgroup analysis based on the tumor grade, treatment, and evaluation criteria, was conducted. The estimated rate of a high NLR was 0.253 (95% confidence interval (CI) 0.198–0.317). The rate of high NLRs was significantly lower in patients with lower tumor grades (G1) than those with higher tumor grades (G2 or G3). In addition, the mean value of the NLR was significantly lower in lower tumor grades than in higher tumor grades. High NLRs were significantly correlated with worse overall and recurrence-free survivals (hazard ratio (HR) 2.180, 95% CI 1.499–3.169 and HR 2.462, 95% CI 1.677–3.615, respectively). In a subgroup analysis, the prognostic implications of the NLR were found in both higher and lower criteria of a high NLR. Taken together, our results show that the NLR could be useful for predicting the tumor grade and the prognosis in PNETs.

## 1. Introduction

Pancreatic neuroendocrine tumors (PNETs) are relatively rare tumors in the pancreas. They account for approximately 1–2% of all pancreatic neoplasms [1]. With the recent advances in imaging modalities and understanding of these tumors, the incidence of PNETs appears to be higher than previously demonstrated. PNETs are categorized as functional or non-functional tumors based on the presence of clinical symptoms. About 70% of PNETs are non-functional and non-functional PNETs are often detected at advanced stages, owing to the absence of specific symptoms [2,3]. Although PNETs have more indolent tumor biology and have a better prognosis than other pancreatic cancers, they are regarded as malignant tumors because of their metastatic potential [4,5]. PNET, a well-differentiated neuroendocrine neoplasm, is composed of cells with nuclear atypia, displaying organoid patterns and lacking necrosis [6]. The tumor grade of PNETs is decided by the proliferative activity, mitosis, and Ki-67 proliferation index. The World Health Organization (WHO) classification system based on mitotic counts and Ki-67 labeling index is useful for predicting the malignant potential of PNETs and the survival rates of patients with PNETs [6]. However, tissue samples for immunohistochemical staining are required to classify the histological grades of these tumors. It is difficult to obtain tumor tissue from some patients. Furthermore, there are no reliable non-invasive markers for evaluating tumor grade and prognosis in patients with PNETs.

Recently, it has been increasingly recognized that tumor progression and patient outcomes are also influenced by the inflammatory response. In particular, the neutrophil-to-lymphocyte ratio (NLR) is helpful in predicting prognosis in cancer patients, as inflammatory responses play critical roles at different stages of tumor development, including initiation, promotion, invasion, and metastasis [7,8]. Several studies demonstrated that the NLR was closely related to the prognosis of various cancers [9,10,11,12]. However, few studies evaluated the prognostic value of the NLR in patients with PNETs. In the present study, we perform a meta-analysis to evaluate the prognostic role of the NLR in patients with PNETs. In addition, the correlation between the NLR and tumor grading of PNETs is investigated to elucidate the predicting role of PNETs for tumor grades.

## 2. Materials and Methods

### 2.1. Published Studies Search and Selection Criteria

The search for the meta-analysis was performed in the PubMed and MEDLINE databases on 31 January 2022. The keywords used were “pancreatic neuroendocrine tumor and neutrophil-to-lymphocyte ratio”. Articles with the correlation between the NLR and survival rate in human PNETs are included in the present study. Case reports or non-original articles were excluded. In addition, the articles written in English were included. In addition, the classification of PNETs was applied by the WHO classification 2017 [6].

### 2.2. Data Extraction

Two authors independently extracted data from eligible studies. The following data were extracted from all the eligible studies: the family name of the first author, year of publication, study location, number of patients analyzed, treatment modality, tumor grade, criteria for a high NLR, and data on the correlations between the NLR and survivals [13,14,15,16,17,18,19,20,21]. For the quantitative aggregation of survival results, the correlation between inflammatory markers and overall survival rate was analyzed using the hazard ratio (HR), such as previous reports [22,23]. The published survival curves were read independently by two independent authors to reduce reading variability. The HRs were combined to yield an overall HR using Peto’s method [24]. This study was performed by Preferred Reporting Items for Systematic Reviews and Meta-Analyses (PRISMA).

### 2.3. Statistical Analyses

In the present meta-analysis, all data were analyzed using the Comprehensive Meta-Analysis software package (Biostat, Englewood, NJ, USA). The rates of high NLRs were investigated according to treatment modality. The correlations between the NLR and overall survival (OS) and recurrence-free survival (RFS) rates were, respectively, evaluated. In the present study, subgroup analysis was conducted based on treatment modality and criteria for high NLRs. The analysis for heterogeneity between the studies was conducted and evaluated using the Q and I^2^ statistics and expressed as *p*-values. Additionally, a sensitivity analysis was conducted to assess the heterogeneity of the eligible studies and the impact of each study on the combined effect. In the assessment of estimated values, because of the eligible studies evaluated in different populations with variable tumor stages and treatments, the application of a random-effect model rather than a fixed-effect model was more suitable. Publication bias was evaluated using Begg’s funnel plot and Egger’s test. If significant publication bias was found, the degree of publication bias was confirmed through fail-safe N and trim-fill tests. The results were considered as statistical significance at *p* < 0.05.

## 3. Results

### 3.1. Selection and Characteristics of the Studies

From the database, 218 reports were identified. Of these, 185 reports were excluded because they were based on other diseases. Seven were excluded because of insufficient or no information. A total of 13 articles were excluded because of duplication. In addition, four articles were excluded due to non-original articles (*n* = 3) and a non-English article (*n* = 1). Finally, nine studies were included in the meta-analysis (Figure 1 and Table 1). Eligible studies included 511 patients with PNETs.

### 3.2. The Higher Neutrophil-to-Lymphocyte Ratio in Pancreatic Neuroendocrine Tumors

The estimated rate of a high NLR was 0.253 (95% CI 0.198–0.317) in the patients with PNETs (Table 2). In a subgroup analysis based on the tumor grade of PNETs, estimated rates of high NLRs were 0.155 (95% CI 0.103–0.227) and 0.419 (95% CI 0.313–0.534) in G1 and G2/3 PNETs, respectively. In addition, estimated rates of high NLRs were 0.367 (95% CI 0.210–0.599) and 0.573 (95% CI 0.395–0.733) in G2 and G3 PNETs, respectively. There were significant differences in the rates of high NLRs between G1 and G2/3 in the meta-regression test. In the present study, the subgroup analysis was conducted based on the criteria for high NLRs. the Subgroups were divided into high and low criteria, according to the median value of the criteria of eligible studies (2.40). The estimated rates of high NLRs were 0.232 (95% CI 0.814–0.287) and 0.390 (95% CI 0.320–0.464) in high and low criteria subgroups, respectively. The mean values of NLRs were 1.912 (95% CI 1.670–2.155), 2.258 (95% CI 1.918–2.598), and 3.363 (95% CI 2.738–3.988) in the grade 1, 2, and 3 subgroups, respectively (Table 3).

### 3.3. Prognostic Implications of the Neutrophil-to-Lymphocyte Ratio in Pancreatic Neuroendocrine Tumors

Furthermore, the correlation between the NLR and survival rate was investigated. Patients with high NLRs showed significantly worse OS and RFS (HR 2.180, 95% CI 1.499–3.169 and HR 2.462, 95% CI 1.677–3.615, respectively; Figure 2 and Table 4). In the subgroup analysis for OS, the HRs of high and low subgroups were 1.978 (95% CI 1.392–2.811) and 4.471 (95% CI 1.531–13.055), respectively. In the subgroup analysis for RFS, the HRs of the high and low subgroups were 2.449 (95% CI 1.416–4.237) and 2.852 (95% CI 1.469–5.537), respectively.

## 4. Discussion

PNETs, which are generally slow-growing tumors, have 33% overall survival rates at 5 years [25]. PNETs are basically divided into non-functioning and functioning tumors. Recently, non-functioning PNETs are increasing and account for >60% in all pancreatic neuroendocrine neoplasms [26,27]. In our meta-analysis, we identified 9 studies that included 1450 PNET patients and investigated the predicting roles of the NLR for the prognosis and tumor grade in patients with PNETs. To the best of our knowledge, our study is the first meta-analysis to evaluate the predicting role of the NLR for tumor grade and prognosis in PNETs.

In summary, the tumor grade of PNETs divides into G1, G2, and G3 by the number of mitoses and the Ki-67 proliferation index. The lower limits of the mitotic count or the Ki-67 proliferation index G3 PNET are 20 mitoses/2 mm^2^ and 20%, respectively [6]. In the present study, the predicting role of the NLR for tumor grade was evaluated. Patients with a G3 PNET had a significantly higher NLR value than those with G1 or G2 PNETs. We analyzed this result through the meta-analysis and meta-regression test. In addition, the estimated rate of a higher NLR was significantly higher in the G3 PNET than G1 or G2 PNETs. If the NLR can predict the tumor grade of PNETs, the NLR is useful for the differentiation of PNETs in cases with the insufficient tissue sample. Although the NLR cannot be completely replaced in grading PNETs, the prediction of the tumor grade is useful in patients with limited samples.

Our results show that high NLRs were correlated with poor OS and RFS in patients with PNETs. Recent studies reported close correlations between high NLRs and poor survivals in other malignancies [9,10,11]. Because a high NLR is associated with a poor prognosis, the NLR can be a candidate for the biomarker in PNETs. The currently valuable prognostic factors, such as the WHO classification, the TNM stage, distant metastases, surgical margin status, tumor sizes, and Ki-67 labeling index, can be evaluated using tumor tissues [28,29]. However, sometimes, it may be difficult to obtain tissue samples. On the other hand, the NLR is inexpensive and may be easily accomplished in daily practice. The NLR value may serve an important function in monitoring the progression of PNETs, as well as predicting patient survivals. Therefore, non-invasive markers that can predict tumor prognosis would be of great importance.

In the eligible studies, various criteria for a high NLR were used from 1.40 to 4.00. In the present study, subgroups were divided into high and low criteria for high NLRs using the median value of the criteria of eligible studies (2.40). The prognostic implications of the NLR were identified in both the high and low criteria subgroups. However, the values of HRs were higher in the low criteria subgroup than in the high criteria subgroup. These patterns were found in both OS and RFS analyses. The estimated rates of the high NLR were 0.390 (95% CI 0.320–0.464) and 0.232 (95% CI 0.184–0.287) in the low and high criteria subgroups, respectively. Individual studies obtained the criteria for a high NLR using the receiver operating characteristic curve. Further cumulative evaluation for the proper criteria of high NLRs is needed.

This study has several limitations. First, the subgroup analysis for the prognostic roles of the NLR based on the tumor grade could not be performed due to insufficient information in the eligible studies. Second, in the present study, the mean values of the NLR were investigated according to the tumor grade. However, the ranges of 95% CI overlapped between G1 and G2 PNETs. The impact of the different measurements of the NLR can be considered. Detailed analysis according to the measurement of the NLR could not be performed due to insufficient information in the eligible studies.

## 5. Conclusions

In conclusion, our meta-analysis demonstrated that the NLR was higher in the high grade than the low grade of PNETs. In addition, the NLR was the predicting factor for the prognosis of patients with PNETs. The NLR can be a candidate for the biomarker in PNETs.

## Figures and Tables

**Figure 1 diagnostics-12-00737-f001:**
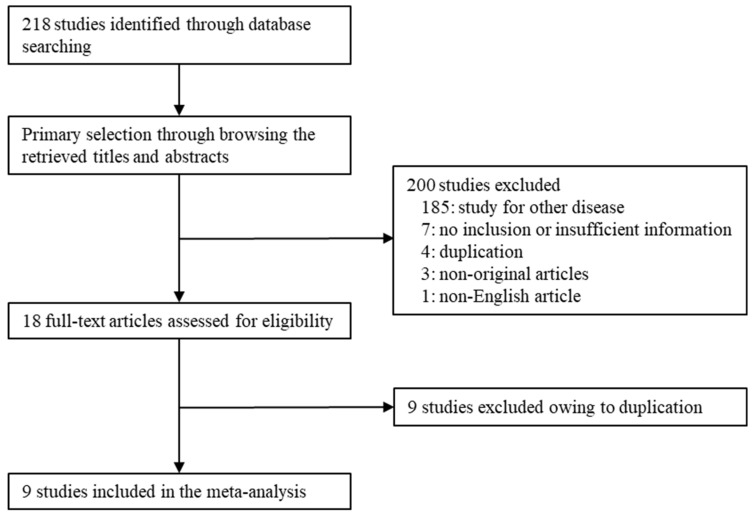
Flow chart of the literature search and selection methods.

**Figure 2 diagnostics-12-00737-f002:**
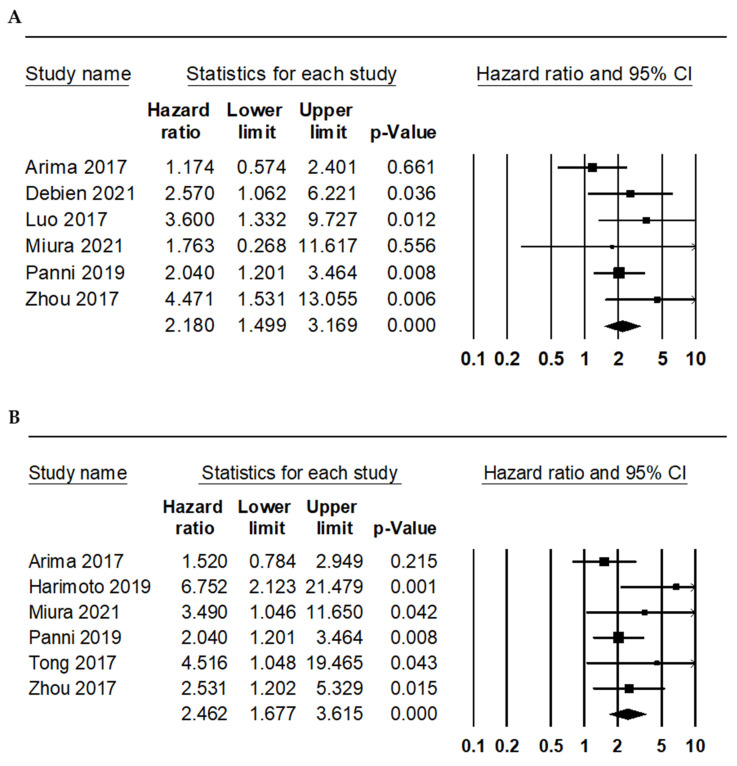
Forest plots for the correlations between a high neutrophil-to-lymphocyte ratio and worse survival rates. (**A**) Overall survival and (**B**) recurrence-free survival.

**Table 1 diagnostics-12-00737-t001:** Main characteristics of the eligible studies.

Author, Year	Location	Tx Option	Tumor Grade			Criteria	Number of Patients		
			G1	G2	G3		Total	High	Low
Arima, 2017	Japan	Surgery	46	9	3	2.40	58	12	46
Debien, 2021	France	Mixed	75	69	0	4.00	144	27	117
Gaitanidis, 2017	U.S.A.	Mixed	ND	ND	ND	2.30	97	ND	ND
Harimoto, 2019	Japan	Surgery	34	17	4	3.41	55	14	41
Luo, 2017	China	Mixed	ND	ND	ND	2.40	89	28	61
Miura, 2021	Japan	Surgery	73	45	2	2.62	120	18	102
Panni, 2019	U.S.A.	Surgery	305	152	20	3.70	620	171	449
Tong, 2017	China	Surgery	52	32	11	1.40	95	ND	ND
Zhou, 2017	China	Surgery	73	76	23	2.31	172	67	105

Tx, treatment; Mixed, surgery with chemoradiation therapy; ND, no description.

**Table 2 diagnostics-12-00737-t002:** Estimated rate of higher value of neutrophil-to-lymphocyte ratio (NLR) in pancreatic neuroendocrine tumors.

	Number of Subset	Fixed Effect [95% CI]	Heterogeneity Test [*p*-Value]	Random Effect [95% CI]	Egger’s Test [*p*-Value]
Overall	7	0.273 [0.249, 0.299]	<0.001	0.253 [0.198, 0.317]	0.404
G1	5	0.168 [0.128, 0.218]	0.098	0.155 [0.103, 0.227]	0.077
G2/3	9	0.408 [0.347, 0.473]	0.031	0.419 [0.313, 0.534]	0.392
G2 *	4	0.392 [0.312, 0.477]	0.011	0.367 [0.210, 0.559]	0.735
G3 ^#^	4	0.573 [0.395, 0.733]	0.579	0.573 [0.395, 0.733]	0.263
NLR criteria					
High (≥2.40)	6	0.253 [0.227, 0.280]	0.017	0.232 [0.184, 0.287]	0.253
Low (<2.40)	1	0.390 [0.320, 0.464]	1.000	0.390 [0.320, 0.464]	-

NLR, neutrophil-to-lymphocyte ratio; CI, confidence interval; *, *p* = 0.007 in the meta-regression test compared to G1; ^#^, *p* < 0.001 in the meta-regression test compared to G1.

**Table 3 diagnostics-12-00737-t003:** Mean value of the neutrophil-to-lymphocyte ratio (NLR) in pancreatic neuroendocrine tumors.

	Number of Subset	Fixed Effect [95% CI]	Heterogeneity Test [*p*-Value]	Random Effect [95% CI]	Egger’s Test [*p*-Value]
Pancreatic neuroendocrine tumor					
G1	3	1.835 [1.719, 1.952]	0.041	1.912 [1.670, 2.155]	0.010
G2/3	5	2.547 [2.318, 2.777]	0.041	2.629 [2.229, 3.028]	0.328
G2 *	2	2.258 [1.918, 2.598]	0.834	2.258 [1.918, 2.598]	-
G3 ^#,†^	2	3.363 [2.738, 3.988]	0.474	3.363 [2.738, 3.988]	-

NLR, neutrophil-to-lymphocyte ratio; CI, confidence interval; *, *p* = 0.117 in the meta-regression test compared to G1; ^#^, *p* < 0.001 in the meta-regression test compared to G1; ^†^, *p* = 0.002 in the meta-regression test compared to G2.

**Table 4 diagnostics-12-00737-t004:** Correlation between the neutrophil-to-lymphocyte ratio (NLR) and survival rate according to the criteria of a high NLR.

	Number of Subset	Fixed Effect [95% CI]	Heterogeneity Test [*p*-Value]	Random Effect [95% CI]	Egger’s Test [*p*-Value]
Overall survival	6	2.141 [1.533, 2.990]	0.325	2.180 [1.499, 3.169]	0.472
NLR criteria					
High (≥2.40)	5	1.978 [1.392, 2.811]	0.433	1.978 [1.392, 2.811]	0.731
Low (<2.40)	1	4.471 [1.531, 13.055]	1.000	4.471 [1.531, 13.055]	-
Recurrence-free survival	6	2.351 [1.701, 3.249]	0.274	2.462 [1.677, 3.615]	0.055
NLR criteria					
High (≥2.40)	4	2.213 [1.528, 3.206]	0.142	2.449 [1.416, 4.237]	0.213
Low (<2.40)	2	2.852 [1.469, 5.537]	0.489	2.852 [1.469, 5.537]	-

NLR, neutrophil-to-lymphocyte ratio; CI, confidence interval.

## Data Availability

Not applicable.

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
