# Peer review of "The Predicting Role of the Neutrophil-to-Lymphocyte Ratio for the Tumor Grade and Prognosis in Pancreatic Neuroendocrine Tumors"

_diagnostics, 2022, doi:10.3390/diagnostics12030737_

Round 1

Reviewer 1 Report

This article entitled with " Predicting roles of neutrophil-to-lymphocyte ratio for the tu-2 mor grade and prognosis in pancreatic neuroendocrine tumors” showed that the NLR could be useful for predicting the tumor grade and the prognosis in PNETs using a meta-analysis.

There are 3 minor points to be revised in this manuscript as below.

<Minor comments>

  • Figure 1 showed ‘9 studies excluded owing to duplication’, but no description found in the text. Please explain the difference from ‘4: duplication’.
  • Which did you use WHO classification 2010 or 2017? Please describe whether neuroendocrine carcinoma was included in G3 PNETs.
  • Have your statistical analyses checked by a statistical expert?

Author Response

This article entitled with " Predicting roles of neutrophil-to-lymphocyte ratio for the tu-2 mor grade and prognosis in pancreatic neuroendocrine tumors” showed that the NLR could be useful for predicting the tumor grade and the prognosis in PNETs using a meta-analysis.

There are 3 minor points to be revised in this manuscript as below.

<Minor comments>

Figure 1 showed ‘9 studies excluded owing to duplication’, but no description found in the text. Please explain the difference from ‘4: duplication’.

Response:

               The number of excluded articles owing to duplication is 13. We corrected the comment in the results of the revised manuscript.

Which did you use WHO classification 2010 or 2017? Please describe whether neuroendocrine carcinoma was included in G3 PNETs.

Response:

               We described based on WHO classification 2017. In addition, in our results, neuroendocrine carcinomas were not included. We added the comment for the classification in the revised manuscript.

Have your statistical analyses checked by a statistical expert?

Response:

               Professor Pyo (co-first author) is an expert in the field who has written 38 meta-analysis papers so far. This study is constructed and analyzed by him.

Reviewer 2 Report

1)Line 29-30. What the Authors mean saying "derived from the endocrine tissue of the pancreas?". Pancreas should be considered as a endocrine gland in its entirety. 

2)Line 168-170 the Authors wrote "Taken together, it may be reasonable to conclude that the NLR may be an ideal prognostic biomarker for PNETs and that a high NLR is associated with a poor prognosis". In my opinion this statement is a little bit unadvised and much more proofs are required to be so drastic about is!

3) Line 178-180. the Authors said " In the present study, subgroups were divided into high and low criteria for high NLR using the median value of the criteria of eligible studies". The Authors should explain better what they mean. 

4) Line 196-197 "In conclusion, our meta-analysis demonstrated that increasing tumor grade of PNET was significantly increased the estimated rate of higher NLR and the value of NLR. " . Also in this case the Authors should explain better what they mean. 

5) Line 197-199 "In addition, the NLR was the predicting factor for the prognosis of patients with PNETs. Clinicians can consider using the NLR as a biomarker in the clinical management of PNETs". Please see point 2.

Author Response

1)Line 29-30. What the Authors mean saying "derived from the endocrine tissue of the pancreas?". Pancreas should be considered as a endocrine gland in its entirety.

Response:

               To avoid the misunderstanding, we corrected the sentence as below:

Pancreatic neuroendocrine tumors (PNETs) are relatively rare tumors that are derived from the endocrine tissue of the pancreas in pancreas.

2)Line 168-170 the Authors wrote "Taken together, it may be reasonable to conclude that the NLR may be an ideal prognostic biomarker for PNETs and that a high NLR is associated with a poor prognosis". In my opinion this statement is a little bit unadvised and much more proofs are required to be so drastic about is!

Response:

               To avoid the misunderstanding, we corrected the sentence as below:

Taken together, it may be reasonable to conclude that the NLR may be an ideal prog-nostic biomarker for PNETs and that Because a high NLR is associated with a poor prognosis, NLR can be a candidate as the biomarker in PNET.

3) Line 178-180. the Authors said " In the present study, subgroups were divided into high and low criteria for high NLR using the median value of the criteria of eligible studies". The Authors should explain better what they mean.

Response:

               Cut-off values of high NLR ranged from 1.4 to 4.0. The real values were 1.4, 2.3, 2.31, 2.4, 2.4, 2.62, 3.41, 3.7, 4.0. The mean value and STDEV of cut-offs were 2.73 and 0.82, respectively. Since the difference between the extremes is not small, the median value was considered to be more suitable than the average.

4) Line 196-197 "In conclusion, our meta-analysis demonstrated that increasing tumor grade of PNET was significantly increased the estimated rate of higher NLR and the value of NLR. " . Also in this case the Authors should explain better what they mean.

Response:

               To avoid the misunderstanding, we corrected the sentence as below:

               In conclusion, our meta-analysis demonstrated that increasing tumor grade of PNET was significantly increased the estimated rate of higher NLR and the value of NLR NLR was higher in the high grade than the low grade of PNET.

5) Line 197-199 "In addition, the NLR was the predicting factor for the prognosis of patients with PNETs. Clinicians can consider using the NLR as a biomarker in the clinical management of PNETs". Please see point 2.

Response:

               To avoid the misunderstanding, we corrected the sentence as below:

Clinicians can consider using the NLR as a biomarker in the clinical management of PNETs NLR can be a candidate as the biomarker in PNET.